# Topological invariance in whiteness optimisation

Johannes S. Haataja [1,2], Gianni Jacucci[1,3], Thomas G. Parton[1], Lukas Schertel [1,4] & Silvia Vignolini [1✉]

Maximizing the scattering of visible light within disordered nano-structured materials is essential for commercial applications such as brighteners, while also testing our fundamental understanding of light-matter interactions. The progress in the research field has been hindered by the lack of understanding how different structural features contribute to the scattering properties. Here we undertake a systematic investigation of light scattering in correlated disordered structures. We demonstrate that the scattering efficiency of disordered systems is mainly determined by topologically invariant features, such as the filling fraction and correlation length, and residual variations are largely accounted by the surface-averaged mean curvature of the systems. Optimal scattering efficiency can thus be obtained from a broad range of disordered structures, especially when structural anisotropy is included as a parameter. These results suggest that any disordered system can be optimised for whiteness and give comparable performance, which has far-reaching consequences for the industrial use of low-index materials for optical scattering.

[1] Yusuf Hamied Department of Chemistry, University of Cambridge, Lensfield Road, Cambridge CB2 1EW, UK. [2] Department of Applied Physics, Aalto University School of Science, P.O. Box 15100, Espoo FI-02150, Finland. [3] Laboratoire Kastler Brossel, ENS-PSL Research University, CNRS, Sorbonne Université, Collège de France, Paris, France. [4] Department of Physics, University of Fribourg, Chemin du Musée 3, 1700 Fribourg, Switzerland. ✉email: sv319@cam.ac.uk

Controlling the transmission of light in optically dense disordered structures is an outstanding scientific challenge[1,2] with relevance to photovoltaic devices, medical imaging[3,4], and random lasing[5,6]. For practical applications such as white pigments and coatings, a common goal is to maximise the reflection of visible light for a given formulation and coating thickness, which is often achieved by creating two-phase disordered structures with large refractive index contrast ($n_2 \gg n_1$). In commercial white paints, the high-index component is typically made from inorganic materials such as titanium dioxide ($TiO_2$, $n = 2.6$)[7], but it is becoming increasingly clear that metal oxide pigments are accompanied by serious concerns for the environment and for human health[8–10]. Although scattering agents made from sustainable and biocompatible sources such as cellulose have emerged as a promising alternative in recent years[11,12], these organic materials are limited to lower refractive index values ($n \lesssim 1.6$). Optimising low-index disordered structures to maximise their scattering efficiency is therefore of growing importance to realise the potential of naturally-sourced white materials.

Scattering optimisation in several species of scarab beetles has resulted in disordered structures with exceptionally bright whiteness, despite the limitation of low refractive index contrast ($n_2/n_1 \approx 1.5$)[13,14]. The high scattering efficiency of these natural photonic materials has been attributed to their anisotropic random network structure[15–17] and has stimulated research into synthetic analogues, with numerous studies reporting comparable optical performance[18–22]. The near-optimal scattering efficiency of these artificial random network structures is notable because their disordered textures are clearly distinct, suggesting that achieving bright whiteness does not require precise mimicry of natural structures. Progress in identifying the universal morphological features that contribute to high scattering efficiency has been hindered by both 1) the observation that theoretical models of light propagation in disordered media are notoriously difficult to solve, arising from the fact that fluctuations of the electric field and permittivity are not statistically-independent, and approximate solutions[23,24] typically assume weak refractive index difference ($(n_2 - n_1)/n_1 \ll 1$), and only consider structural correlations up to second-order[25], and 2) by an absence of systematic numerical investigation of disorder structures, with previous studies typically focused on individual models such as random sphere packing[26] or spinodal decomposition[21]. Furthermore, comparison between different types of random structure requires robust methods to quantify and classify disordered morphologies, which have previously been lacking.

In this work, we use in silico synthesis to generate a comprehensive range of two-phase nanostructures with correlated disorder and simulate their optical response across the visible range as illustrated by the workflow in Fig. 1. We find that the scattering efficiency is largely determined by the first- and second-order statistical properties of the system, captured by the filling fraction and two-point correlation function, respectively, even for structures with large refractive index contrast where higher-order effects are expected to be significant, and that the optimal value for the filling fraction is determined by the refractive index contrast regardless of the disorder morphology. Our morphological analysis using Minkowski functionals reveals that the residual variation in scattering efficiency between different structural classes can be attributed to specific topological properties of the structure, such as integral mean curvature and surface area. We then show that the scattering efficiency of all structural classes can be further enhanced by introducing structural anisotropy as an additional tuning parameter. These results indicate that near-optimal scattering efficiency can readily be achieved with any kind of disordered structure by tuning a handful of morphological parameters.

## Results and discussion

**Structure generation and morphological analysis.** The salient morphological features of disordered two-phase structures, $I(\mathbf{x}) : \mathbb{R}^3 \to \{0, 1\}$, are often captured by a few low-order ensemble properties. A common approach to describing ensemble structural properties is to use $n$-point correlation functions $S_n = \langle I(\mathbf{x}_1)I(\mathbf{x}_2)\dots I(\mathbf{x}_n) \rangle$, where angular brackets denote ensemble averaging. The 1-point correlation $S_1 = \langle I(\mathbf{x}) \rangle$ is equal to the filling fraction of the high-index phase ($\phi = V_0/V \in [0, 1]$ where $V$ is the total volume of the two phases). The 2-point correlation, as a function of the distance $r = |\mathbf{r}|$, is given by

$$S_2(r) = \langle I(\mathbf{x})I(\mathbf{x} + \mathbf{r}) \rangle \qquad (1)$$

and decays from a maximum value of $S_2(0) = \phi$ at $r = 0$ to $S_2 \to \phi^2$ as $r \to \infty$[27]. Between these limits, the variation of $S_2(r)$ for correlated disordered structures is typically sinc-like, but in some cases decays monotonically without oscillation. To accommodate characteristic length scale in both cases, we approximate the non-oscillating decay in $S_2(r)$ using Corson's formula[28] and define the correlation length $l_c$ as a fixed intensity cut-off (see Eqs. (19) and (20) in the Methods -section, and Supplementary Note 1 in Electronic Supporting information (ESI)). Previous studies on natural and synthetic white materials have established that near-optimal scattering efficiency is achieved by specific random structures with correlated disorder on a length-scale comparable to the wavelength of visible light, so we focused on structures with correlated disorder in the range $l_c \in [100, 900]$nm for optimisation investigation[16,18,21,22].

To understand the role of fine structural features in the scattering efficiency, we set out to produce the widest possible range of two-phasic correlated disordered structures, $I(\mathbf{x})$, obtained from simulated scalar fields $f(\mathbf{x}) : \mathbb{R}^3 \to \mathbb{R}$ using a threshold scheme (see Methods, Eq. (10)), that shared lower-order statistical features, i.e. filling fraction $\phi$ and correlation length $l_c$, going beyond previous studies on multiple scattering in specific types of structures such as randomly packed spheres or bicontinuous structures[20–22].

We first generated two structural classes using Gaussian random fields, also known as Gaussian processes (GPs), which are uniquely determined by their first and second order statistics[29–31]. Structures generated by GPs provide an essential comparison with other classes of disordered structures, as the latter allow us to isolate the role of finer morphological features on the optical response. We generated scalar fields $f(\mathbf{x}) = GP(m(\mathbf{x}), K(r))$ using GPs with zero mean $m(\mathbf{x}) = \langle f(\mathbf{x}) \rangle = 0$, and covariance (kernel) functions $K(r) = \langle f(\mathbf{x})f(\mathbf{x} + \mathbf{r}) \rangle$ (c.f. Eq. (11)). The two classes of GP structures were generated with sinc-like or squared-exponential kernel functions:

$$\begin{aligned} \text{GP1: } K(r) &\propto \frac{\sin(l_1 r)}{l_1 r}, \\ \text{GP2: } K(r) &\propto \exp\left(\frac{-r^2}{2l_2^2}\right) \end{aligned} \qquad (2)$$

where $l_1, l_2$ are scaling parameters. Note that the $S_2(r)$ is analogous, but not equal to $K(r)$ due to the thresholding scheme used to create the binary field $I(\mathbf{x})$[32,33].

Previous experimental studies have often generated two-phase disordered structures by spinodal decomposition (SD)[21], the spontaneous demixing of mobile phases in the absence of a thermodynamic barrier. We therefore generated three structural classes using SD as a comparison with the GP structural classes. The evolution of a two-phase system undergoing spinodal decomposition is well-described by the Cahn–Hilliard (CH)

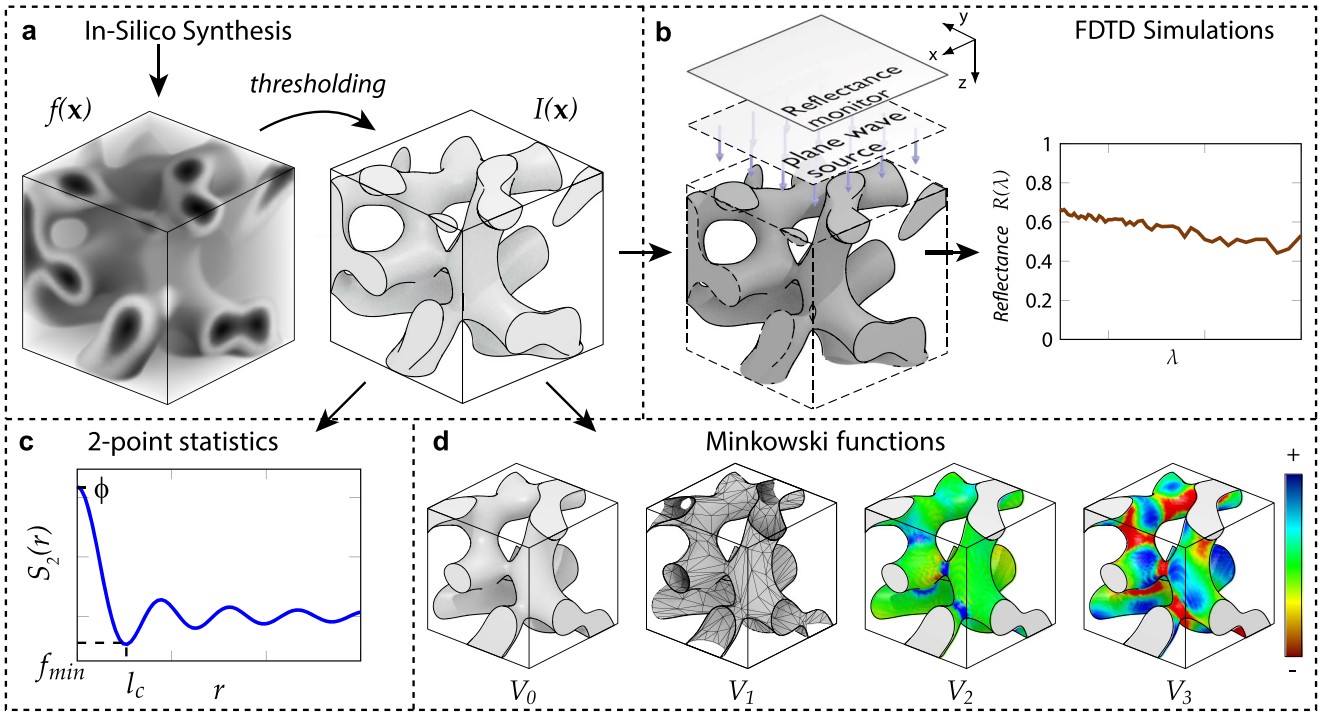

**Fig. 1 Workflow of structure generation, optical simulations and morphological analysis. a** In silico synthesis results in an inhomogeneous disordered structure described by scalar field $f(\mathbf{x})$, which is then thresholded to produce a two-phase structure. **b** Illustration of finite-difference time-domain (FDTD) optical simulations performed with broadband plane-wave illumination along the $z$ axis to obtain total reflectance spectra $R(\lambda)$. **c** The 2-point correlation function $S_2(r)$, with the filling fraction $\phi = V_0/V$ and correlation length $l_c$ indicated. **d** Illustration of the piece-wise contributions to the scalar Minkowski functionals: volume $V_0$, $V_1$ ($\propto$surface area), integral mean curvature $V_2$ and integral Gaussian curvature $V_3$.

phase-field model, which provides a dynamic equation for the evolution of a scalar field $f(\mathbf{x})$ due to free energy minimisation[34]. We generated three structures by simulating the evolution of a scalar field obeying the CH Eq. (12), initialised using 50%, (SD1), 30%, (SD2), and 70%, (SD3) filling fractions. Note that the scalar fields used to generate SD2 and SD3 were complementary ($f_{SD2}(\mathbf{x}) = -f_{SD3}(\mathbf{x})$), but the resulting structures have identical filling fractions due to different choices for the threshold value $\rho_0$.

Although the conventional CH model has been successfully used to simulate disordered structures in several fields, it does not provide access to the full gamut of correlated disordered structures. Specifically, the CH equation results in bi-continuous networks or isolated droplet-like structures, but cannot generate other morphological motifs, such as tubular segments or cell-like structures, which are commonly observed in experimental systems. To access a broader range of disordered morphologies, we generated structures using the Functionalized Cahn–Hilliard (FCH) model (Supplementary Eqs. (S6) and (S7)). The FCH model extends the conventional CH model by including free energy terms to account for the effects of mixing entropy and hydrophobic interactions, which determine the curvature of the interfaces between the two phases[35–37]. We generated five further types of correlated disordered structures by adopting a previously reported protocol using the FCH model with 20 % initial filling fraction[35].

As intended, the simulated structures capture numerous morphological motifs observed in physical disordered structures, including isolated "colloidal" structures (e.g. FC1), branched networks (e.g. GP1, SD1), tubular inclusions (FC2-FC3) and cellular structures (e.g. FC5).

Although structures in Fig. 2a are easily distinguishable by visual inspection, they are almost indistinguishable in terms of 2-point correlation function, as shown in Fig. 2b, for fixed filling fraction ($\phi = V_0/V = S_2(0) = 30\%$) and correlation length ($l_c =$

300 nm). This similarity in lower-order statistics was achieved by iterative refinement of the in silico synthesis conditions to converge on the desired properties. These $\phi$ and $l_c$ values were chosen as typical near-optimal values, comparable to those observed in natural systems with similar refractive index contrast[38].

The fine morphological features that distinguish the structures are encoded in higher-order $n$-point correlation functions ($S_n$ for $n > 2$), but unlike the intuitive properties that arise from the lower-order correlations (i.e. $\phi$ and $l_c$), these higher-order correlations are more difficult to interpret in terms of salient morphological features. Moreover, the higher-order correlations are more computationally intensive to calculate and visualise, as the output of each $S_n$ is a function on a multidimensional space $\mathbb{R}^{3(n-1)}$. As an alternative to $n$-point correlation functions, we used scalar Minkowski functionals $V_j$ to quantitatively discriminate between disordered structures. These translation and rotation invariant functionals are obtained by surface or volume integrals over the 3D structure, and are given by

$$V_0(I) = \int_I dV,$$ (3)

$$V_1(I) = \frac{1}{3} \int_{\partial I} ds$$ (4)

$$V_2(I) = \frac{1}{6} \int_{\partial I} \frac{1}{2} \left( \frac{1}{r_1(s)} + \frac{1}{r_2(s)} \right) ds,$$ (5)

$$V_3(I) = \frac{1}{3} \int_{\partial I} \frac{1}{r_1(s) r_2(s)} ds$$ (6)

The functionals correspond to the volume of the high-index phase $V_0$, total surface area $V_1$, integral mean curvature $V_2$ and

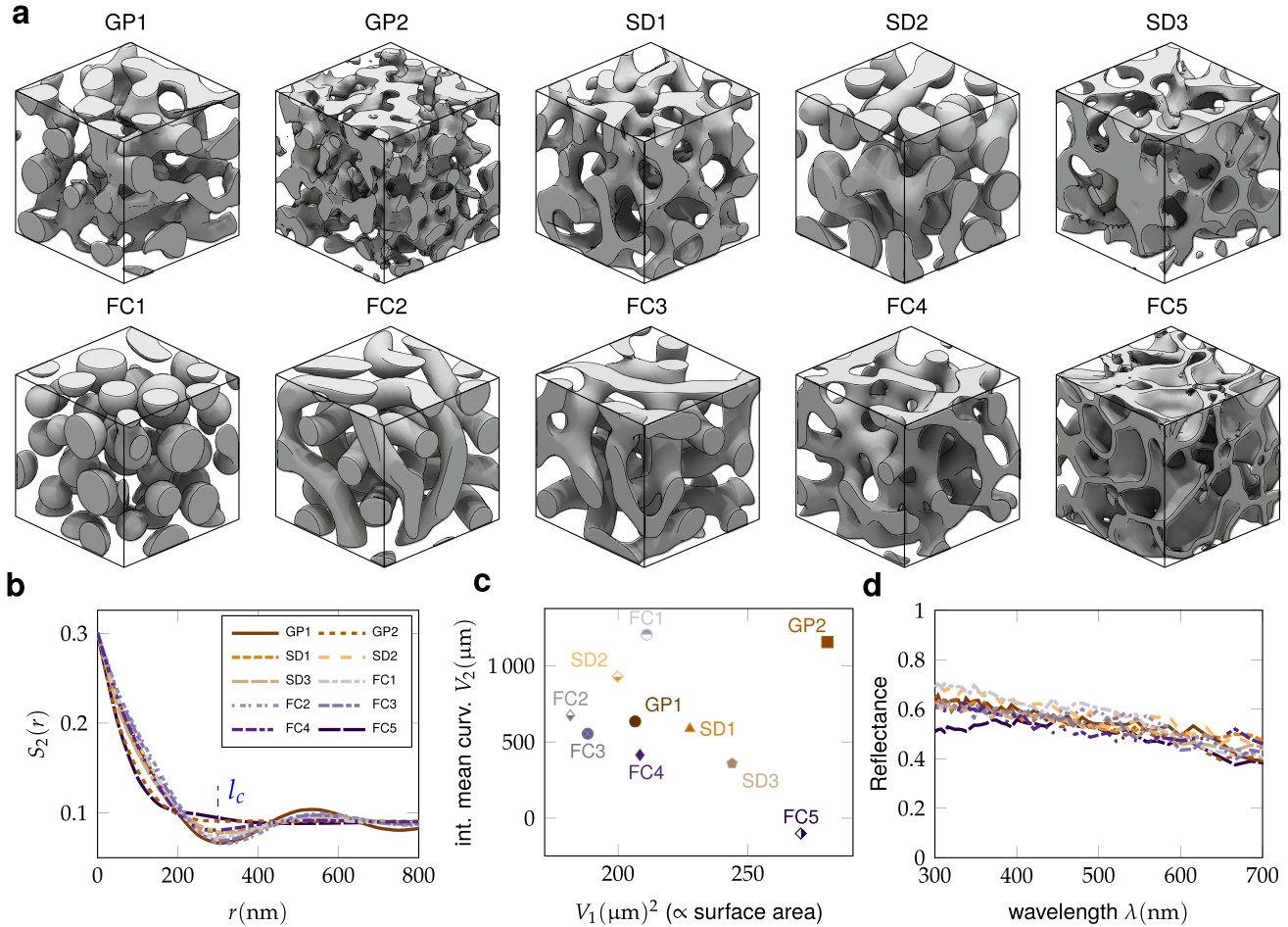

**Fig. 2 Different morphologies, their quantitative descriptors and optical invariance. a** Examples of each class of two-phase disordered structure produced by in silico synthesis for $\phi = 30\%$, $l_c = 300$ nm: structures generated by Gaussian Processes (GP*), spinodal decomposition (SD*) using the simple Cahn–Hilliard model, and Functionalized Cahn–Hilliard models (FC*). **b** Two-point correlation function $S_2(r)$ for the different structures are nearly identical, whereas **c** using $V_1$ (∝surface area), and integral mean curvature $V_2$ the structures are distinguishable. **d** Reflectance spectra of the structures in **a**, showing near-identical scattering across the visible range.

integral Gaussian curvature $V_3$ respectively, as illustrated in Fig. 1d.

Plotting the structures in the $V_1 - V_2$ parameter space (Fig. 2c), we can readily distinguish between different structural classes by their total surface area and mean curvature. For instance, we observe that the 'cellular' structure FC5 has the largest surface area ($V_1$) and negative mean curvature ($V_2 < 0$) while $V_2 > 0$ all other structures. The information encoded in the scalar Minkowski functionals also manifests in the 2-point correlation function. For instance, in the limit of small correlation distance $r$ it can be shown that

$$S_2(r) \approx \frac{V_0}{V} - \frac{3}{4}\frac{V_1}{V}r + \mathcal{O}(r^3), \tag{7}$$

with higher-order terms related to the mean principal curvatures[39,40]. Comparing the $V_1$ values in Fig. 2c with 2b, we see that the gradient of the initial decay of 2-point correlation function with $r$ is proportional to the surface area (most prominently for high-$V_1$ classes such as FC5 and GP2), consistent with Eq. (7). The scalar Minkowski functionals thus provide a computationally inexpensive method to extract useful morphological features with intuitive geometric interpretations.

**Insensitivity of the reflectance to higher-order morphological features.** Surprisingly, the simulated reflectance spectra for the

ten structural classes are highly similar (Fig. 2d), despite their clear morphological differences. All structural classes showed slightly higher reflectance at the blue end of the spectrum, in agreement with previous reports[18,21]. In terms of the spectrally-averaged mean reflectance, we found <10% variation between the highest and lowest values (corresponding to the 'spherical' structure FC1 and 'cellular' structure FC5 respectively). This insensitivity of the reflectance spectrum to fine morphological features suggests that the statistical properties up to second order ($\phi = 30\%$, $l_c = 300$ nm) account for most of the scattering efficiency of these isotropic correlated disordered structures.

To understand whether this invariance in scattering efficiency was a universal feature of correlated disordered structures, we systematically explored the $\phi - l_c$ parameter space with additional FDTD simulations. For each structural class, we performed line sweeps by keeping $l_c$ fixed and varying $\phi$ (or vice versa), and calculated the mean reflectance for each structure, as summarised in Fig. 3a–c. Calculations were performed in the range $\phi(\%) \in [10, 60]$ and $l_c\,(\text{nm}) \in [100, 900]$ to encompass the optimal parameter range.

For relatively low correlation lengths (e.g. $l_c = 300$ nm), the total reflectance is highly similar between all structural classes, with the notable exception of the 'cellular' FC5 class at low filling fraction (Fig. 3a). Apart from FC5, all structural classes showed unimodal variation in $R_{\text{tot}}$ versus filling fraction, with a broad

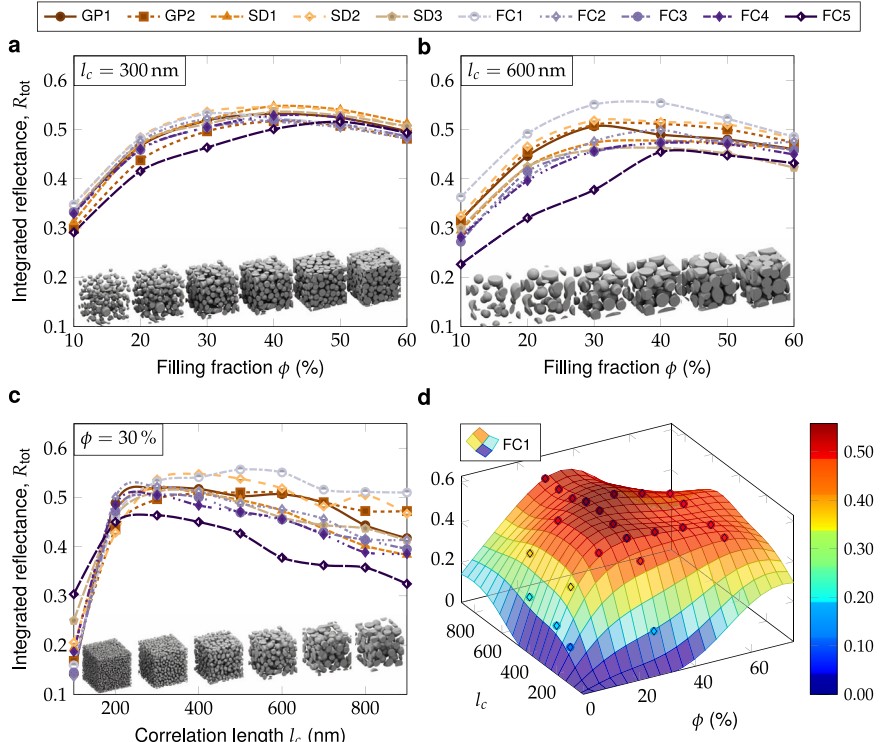

**Fig. 3 Line searches along different filling fractions $\phi$ and correlation lengths $l_c$ for different structural classes. a, b** Varying filling fraction with fixed $l_c = 300$ nm and $l_c = 600$ nm. **c** Varying correlation length with fixed $\phi = 30\%$. Cubical volume snapshots at the bottom in **a–c** illustrate the effect of changing the filling fraction an the correlation length respectively. **d** Extrapolated surface for the structural class with highest average reflectance (FC1).

optimal range around $\phi \approx 35\%$, in agreement with previous studies[20,21,41]. At larger correlation lengths, greater variability in reflectance values between structural classes was observed, as exemplified in Fig. 3b for $l_c = 600$ nm, with the optimal filling fractions again located around 35%. The best-performing structural classes were those with more 'colloidal'-type morphological motifs (FC1, SD2), while the worst-performing were more 'cellular' (SD3, FC5), with performance loosely correlated with mean curvature (cf. Fig. 2).

These trends become more apparent when comparing reflectance values at $\phi = 30\%$ for a range of correlation lengths, as shown in Fig. 3c. At very low correlation lengths ($l_c < 300$ nm), the reflectance for all structural classes is much lower and decreases sharply with increasing wavelength. This behaviour is expected as the correlation length approaches the sub-wavelength regime (cf. Supplementary Fig. S5). The variation in reflectance between structural classes increases with correlation length, though remaining < 20% even for the largest $l_c$ values. There is no clear optimal correlation length, as the total reflectance is more or less constant across the 300 nm to 600 nm range, and the correlation length at maximum reflectance varies considerably between structural classes. Across all correlation lengths, the structural classes generated by Gaussian processes (GP1-2) (that lack higher-order morphological features) always have intermediate reflectance values, while the 'colloidal' structural classes out-perform the others, and the 'cellular' structural classes perform especially poorly.

We interpolated the line scans to produce a 3D plot of the total reflectance in the $\phi - l_c$ parameter space, as shown in Fig. 3d for the best-performing FC1 structural class. The broad plateau of optimal reflectance originates from the spectral averaging of $R_{tot}$ across the visible range (see Methods), which smooths out peaks and shoulders originating from Mie resonances[42].

The similarity between the reflectance spectra in Fig. 3a–c indicates that second-order statistical properties of two-phase disordered structures are sufficient to explain the most of the scattering response, and near-optimal broadband reflectance can be achieved robustly from a fairly wide range of $\phi$ and $l_c$ values. This conclusion is notable, as it has previously been suggested that structures with near-identical reflectance spectra possess unique morphological similarities[21]. Our results suggest that, on the contrary, the reflectance spectrum is mainly predicted by the filling fraction and correlation length, and a broad variety of structures map onto near-identical optical responses.

We considered these results in the context of multiple scattering theory for optically dense random structures (see Supplementary Note 2). In this formalism, the scattering from an inhomogeneous structure is expressed as an perturbative expansion, with the relative contribution of higher-order terms dependent on the magnitude of the refractive index contrast[25]. In the limit of weak contrast ($(n_2 - n_1)/n_1 \ll 1$), the total scattering response can be accurately approximated by truncating the expansion at second order. This approximation leads to expressions for the scattering and transport mean free paths that depend only on the 2-point correlation function. This second-order description (also known as the bilocal approximation) is also frequently applied beyond the low-contrast limit as the higher-order statistical properties of experimental disordered structures are difficult to determine with sufficient accuracy[25]. In our case, where the refractive index contrast is moderate ($n_2/n_1 = 1.5$), the second-order statistical description appears to account for the optical response.

To investigate the role of refractive index contrast, we repeated the $\phi = 10\%$ to 60% line scans in Fig. 3a for $l_c = 300$ nm and $n_1 = 1$, while varying $n_2 = \{1.1, 1.5, 2.6\}$. In the low-contrast case ($n_2/n_1 = 1.1$, Supplementary Fig. S2a, the reflectance curves for all structural classes are similar, and exhibit an optimal value near 50% filling fraction. With increasing contrast, the optimal filling fraction decreases substantially, and is accompanied by a

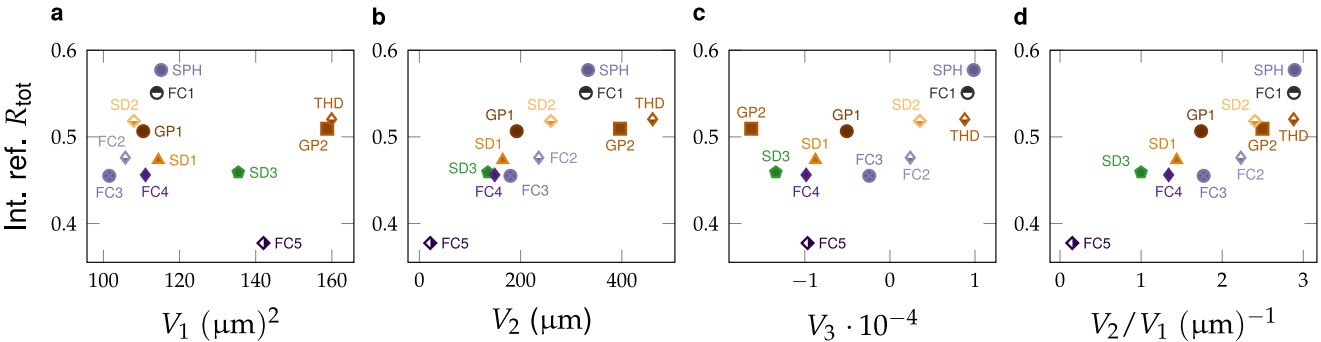

**Fig. 4 Scattering dependence on Minkowski functions.** Total reflectance, $R_{tot}$, vs. **a** $V_1$, **b** $V_2$, **c** $V_3$ and **d** the ratio $V_2/V_1$ for $\phi = 30\%$ and $l_c = 600$ nm. As shown in **d** the differences in $R_{tot}$ between different structures are well correlated with the surface-averaged mean curvature value $V_2/V_1$. (c.f. Supplementary Fig. S7 to see plots for full range of correlation lengths). For New labels SPH and THD denote random assemblies of spheres and tetrahedra respectively.

significant increase in the magnitude of total reflectance (cf. Supplementary Fig. S2a–c). These findings are consistent with previous studies of random assemblies of spheres[26] and can be interpreted as a greater effective filling fraction of the high-index phase due to near-field coupling[43,44].

For each given value of $n_2/n_1$, there is relatively little variation in the optimal filling fraction and maximum reflectance between different structural classes, indicating that these values are primarily constrained by $n_2/n_1$. By considering the optical response in both the independent scattering and bilocal approximation, the optimal filling fraction can be estimated using Supplementary Eq. (S6) and Supplementary Fig. S4, respectively, see ESI. In both cases the optimal filling fraction is found to decrease with $n_2/n_1$ in good agreement with the simulation results. These results confirm that for low-contrast random media, a higher loading of scattering elements is required to achieve optimal reflectance.

**The role of interfacial curvature.** Having established that all structural classes give comparable, but not identical, reflectance values for fixed $\phi$ and $l_c$, we then considered which specific morphological features account for the residual differences in optical response (see Supplementary Note 3). Surveying the results of the line scans in Fig. 3a–c, we observed that FC1, SD2 and GP2 are frequently the best-performing structural classes for a range of $\phi$ and $l_c$ values, while the worst-performing were frequently FC5, FC4 or SD3. We therefore examined the Minkowski functionals for structures from each class at $\phi = 30\%$, as illustrated in Fig. 4a–c (for $l_c = 600$ nm (see Supplementary Fig. S7 for data for all $l_c$ values). The curvature of the interface between the two phases clearly plays an important role: the reflectance is strongly positively correlated with integral mean curvature $V_2$ (Fig. 4b), and is also weakly correlated with the integral Gaussian curvature $V_3$ (Fig. 4c). Interestingly, the reflectance is tightly correlated with surface area $V_1$ for many structural classes, but high-$V_1$ outliers such as FC5 and GP2 have widely varying values (Fig. 4a). We note that substantially different trends are observed for low $l_c$ (Supplementary Fig. S7), but the reflectance spectra in this regime are strongly influenced by Mie resonances at short wavelengths (cf. Supplementary Fig. S5), which complicates interpretation of the data.

To better understand the role of curvature, we introduced random assemblies of monodisperse spheres (SPH) and tetrahedra (THD) as two supplementary structural classes, with structures generated by molecular dynamics simulations of hard particle interactions (see Supplementary Note 4 and Supplementary Fig. S8). These additional structural classes allow us to tune the morphology of the scattering agents more directly. As shown

in Figs. 4 and Supplementary Fig. S8, the reflectance from SPH structures was consistently among the highest of all structural classes and close to values from the morphologically similar FC1 'colloidal' structures. However, the THD structures, which had much greater $V_1$ values but only slightly higher $V_2$ values than the SPH structures, exhibited good but unexceptional scattering performance. These results indicate that optimal performance arises from a balance between increasing the integral mean curvature without incurring the penalty of excessive surface area.

Based on the behaviour of the 12 structural classes, we concluded that the residual variation in reflectance was largely explained by the value of the ratio $V_2/V_1$, which we describe as the 'surface-averaged mean curvature' (i.e. the integrated mean curvature value averaged over the interface area between the two phases). The surface-averaged mean curvature is strongly correlated with the reflectance for a broad range of correlation lengths $l_c \geq 400$ nm (Fig. 4d, Supplementary Fig. S4) and therefore provides a universal morphological parameter to explain the differences in performance between structural classes.

The 'colloidal' structural classes, such as SPH and FC1, have the highest $V_2/V_1$ values. The drawback is that this type of structure, with isolated spheroidal domains of the high-index phase surrounded by a low-index matrix, is difficult to produce in physical disordered systems. In particular, producing 'colloidal' domains in a low-index matrix is not practical in biological disordered structures, where the low-index phase is required to be air ($n_1 = 1$) to maximise refractive index contrast. As a consequence, the highly scattering random networks observed in *Cyphochilus spp.* and *L. stigma* have the additional constraint of forming continuous freestanding structures.

**Structural anisotropy as an additional optimisation parameter.** Despite being limited by the requirement that the high-index phase is a connected continuous network, beetle scales are able to out-perform the 'network' type structural classes described above using anisotropic disordered structures[17]. Structural anisotropy has previously been shown to enhance the scattering efficiency of correlated disordered media relative to isotropic structures of the same filling fraction and therefore provides an additional parameter for whiteness optimisation[41].

To understand the influence of anisotropy on scattering efficiency, robust and universal measures of structural anisotropy are required. A popular way to quantify structural anisotropy is to determine the 2D correlation lengths along each sample axis (e.g. $\mathbf{l}_c = [l_{c,x}, l_{c,y}, l_{c,z}]$) and consider their ratios (e.g. $l_{c,x}/l_{c,z}$)[15,21,45–48]. Anisotropy can also be determined from the scattering response in terms of the ratios of the transport mean free paths along

different axes[17]. However, the proper use and interpretation of these values remains an open question. Here we quantify structural anisotropy using the so-called Minkowski tensors (tensorial Minkowski functionals), which have previously been applied to other anisotropic systems[49,50]. For an anisotropic two-phase medium described by the indicator field $I(\mathbf{x})$, the rank-2 Minkowski tensor $W_1^{0,2}$ is particularly relevant, as it describes the angular distribution of surface normals. This tensor is defined as

$$W_1^{0,2}(I(\mathbf{x})) := \frac{1}{3} \int_{S^2} \rho(\mathbf{n})\, \mathbf{n} \otimes \mathbf{n}\, d\Omega \qquad (8)$$

where $S^2$ is the unit sphere and $\rho(\mathbf{n})$ is the angular distribution of surface normals $\mathbf{n}(\theta, \phi)$ across all solid angles $\Omega$[49,51]. We can then define an anisotropy parameter $\alpha$ in terms of the maximum and minimum eigenvalues of $W_1^{0,2}$ as follows:

$$\alpha := \frac{|\mu_{\max}| - |\mu_{\min}|}{|\mu_{\max}|} = 1 - \frac{|\mu_{\min}|}{|\mu_{\max}|} \in [0, 1]. \qquad (9)$$

For instance, an isotropic structure has a spherically symmetric surface normal distribution ($\rho = 1/4\pi$) so $\mu_{\min} = \mu_{\max}$ and $\alpha = 0$ as expected. The parameter $\alpha$ thus provides a simple measure of structural anisotropy that can be directly connected to the morphological properties of the two-phase medium.

To compare this parameter to previous measures of structural anisotropy, we calculated $\alpha$ for beetles scales using publicly available X-ray tomography datasets[38]. We find that $\alpha = 0.4$ and $\alpha = 0.3$ for beetle scales from *Cyphochilus spp.* and *L. stigma* respectively. The substantial anisotropy in the latter case is especially noteworthy, as it has been suggested that the *L. stigma* structure is in fact isotropic, based on 2D correlation length analysis along different axes[21]. Our finding is consistent with the original claims of anisotropy in this structure[16,17], and demonstrates that $\alpha$ captures latent anisotropy not evident from other metrics.

To explore the role of anisotropy as an additional optimisation parameter, we expanded our investigation to include anisotropic variants of the ten structural classes introduced above (see Supplementary Note 5). We generated structures with uniaxial anisotropy parallel to the light propagation direction by several methods. For the GP structural classes, anisotropy was achieved by scaling the kernel function, while for the the SD and FC structural classes, anisotropy was introduced into the free energy of the scalar field $f(\mathbf{x})$ (see methods and Eq. (17) for further details).

We simulated the total reflectance for each structural class across a range of $\alpha$ values while keeping the filling fraction and correlation length constant ($\phi = 30\%$, $l_c = 300$ nm), as shown in Fig. 5. In the limit of low anisotropy, the reflectances tend to the values found previously for isotropic structures (i.e. the $\phi = 30\%$ data points in Fig. 3a). The maximum reflectance ($R_{\text{tot}} \approx 0.6$) typically occurred within the range $\alpha \in [0.4, 0.6]$, but the optimum anisotropy value varied considerably between structural classes. The observation that optimal performance occurs at intermediate anisotropy values is consistent with a previous study on anisotropic 2D systems[41]. Almost all structural classes exhibited a sharp decrease in reflectance in the limit of high anisotropy ($\alpha \approx 1$), where the structures become quasi-1D multi-layer systems and can therefore exhibit intense but narrow reflectance peaks due to localised modes[52]. The exception was the 'cellular' FC5 structural class, whose structures showed increased reflectance at high anisotropy (Fig. 5). Surprisingly, FC5 structures achieved the highest overall reflectance of any structural class at 30 % filling fraction ($R_{\text{tot}} \approx 0.63$) for $l_c = 200$ nm, $\alpha \approx 0.85$, as shown in figure S9. These results show that further optimisation of scattering efficiency is possible when anisotropy is included as

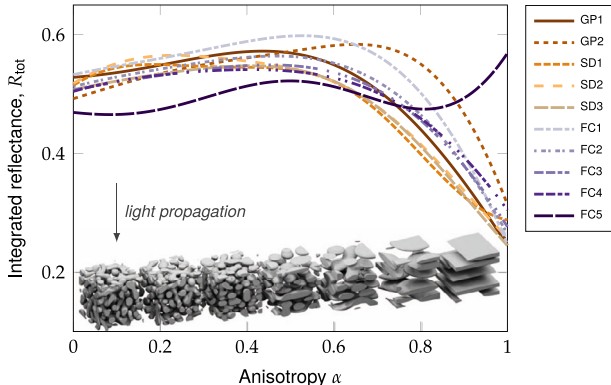

**Fig. 5 Effect of anisotropy.** The average reflectance vs. structural anisotropy value $\alpha$ for each structural class with $\phi = 30\%$ and $l_c = 300$ nm. Interpolations of the simulated data are shown for clarity (see Supplementary Fig. S10 for data points). The inset figures show examples of FC1 structures with increasing anisotropy.

a parameter, and even relatively poor reflectance can be greatly optimised using structural anisotropy.

To further understand the influence of anisotropy on the scattering efficiency and compare our results with previous studies, we applied the diffusion approximation to estimate the anisotropic scattering and transport mean free paths for the anisotropic disordered structures used in Fig. 5. Although the total reflectance $R_{\text{tot}}$ shown in Fig. 5 and previous figures is a useful measure of the overall efficacy of scattering materials for practical applications, it depends on the thickness of the structure (5 μm in this case), making it difficult to make comparisons between studies. In contrast, the scattering and transport mean free paths ($l_s$ and $l_t$ respectively) are intrinsic measures of the scattering efficiency.

In general the diffusion approximation is valid[53,54] for isotropic structures with optical thickness $L/l_t \geq 8$. As Lee and coworkers note[45] the validity also depends on the nature of the scattering[55–57] and the thickness limit is lowered as the index contrast decreases[58,59]. Therefore we estimated the $zz$-components of the mean anisotropic mean freepath tensors, $l_{s,zz}$ and $l_{t,zz}$, based on the anisotropic diffusion theory[45,60,61] (see Supplementary Eq. (S19) and (S20)), by measuring the simulated ballistic and total transmittance while varying the sample thickness (see Supplementary Note 6), and the validity of the diffusion approximation was confirmed with an additional simulation (cf. Supplementary Fig. S12). The anisotropic transport mean free path is found to increase sharply at high $\alpha$ for all structural classes, with the exception of FC5 (Supplementary Fig. S13) while the scattering mean free path remains more constant (Supplementary Fig. S14). The divergence between $l_{s,zz}$ and $l_{t,zz}$ can be attributed to a growing anisotropy factor in the scattering angle, with forward scattering more strongly favoured at high $\alpha$. For most structural classes, the minima in $l_{t,zz}$ at intermediate $\alpha$ values (indicating optimal scattering efficiency) are accompanied by a convergence in $l_{t,zz}$ values across the visible wavelength range, indicating a more uniformly white transmission spectrum. Structural anisotropy therefore has the additional benefit of improving the whiteness of disordered materials, as previously noted[41].

## Conclusions

In this work, we aimed to elucidate the connection between morphology and scattering efficiency in two-phase disordered structures. We developed a novel in silico workflow to generate and characterise random structures of a desired type and texture. This workflow can readily be adapted to generate disordered

structures for diverse applications in computational materials science[62]. In our case, we produced a comprehensive range of disordered morphologies for FDTD simulations to investigate how light scattering is influenced by several fundamental morphological parameters.

First, we demonstrated that the optimal filling fraction $\phi$ lay around 35% for all structural classes (with that value primarily dependent on the refractive index contrast $n_2/n_1 = 1.5$), and the optimal correlation length lay in the visible range ($l_c \approx 500$ nm). Our findings justify why previous studies give similar trends when seeking to optimise the scattering efficiency[26,41]. We then demonstrated that if $\phi$ and $l_c$ are fixed, the remaining variation between isotropic structural classes is mostly explained by the surface-averaged mean curvature $V_2/V_1$. Finally, we explored the influence of structural anisotropy, and found that it allowed further enhancement of the scattering efficiency, with optimal anisotropy typically occurring in the $\alpha = 0.4$–$0.6$ range.

Our results suggest that the type of disordered morphology (i.e. colloidal, network, cellular) chosen to create a white material is ultimately irrelevant, as any structural class can give comparable scattering efficiency after optimisation. From an industrial perspective this conclusion is fortuitous, as optimal whiteness can be achieved without closely imitating a particular type of disordered structure. The universality of optimal whiteness also explains why there are many different disordered structures in biological systems[14,63–71], as there is not a strong evolutionary pressure to converge on a single structural class. However, we expect morphology to play a greater role in structures with higher refractive index contrast, where higher order interactions from multiple scattering events are more significant. We hope that our observations provide significant insights for deriving analytical expression for light propagation in disordered systems with correlation length $l_c$ and $\lambda$ on the visible range, which have been so far difficult to obtain[25].

In summary, there is no unique route to brilliant whiteness in two-phase disordered structures.

## Methods

All the in silico syntheses were carried out in a $N \times N \times N$ cubic grid with periodic boundary conditions, and set to correspond to a $L^3 = 5\,\mu\text{m} \times 5\,\mu\text{m} \times 5\,\mu\text{m}$ box. A value of $N = 100$ was used for all structures with $l_c \geq 200$ nm, and $N = 200$ otherwise.

**Conversion to binary fields.** The stochastic fields $f(\mathbf{x}) : \mathbb{R}^3 \rightarrow \mathbb{R}$ synthesised with the different methods were converted to the two phase disordered structures $I(\mathbf{x}) : \mathbb{R}^3 \rightarrow \{0, 1\}$ using a simple thresholding scheme

$$I(\mathbf{x}) = \begin{cases} 0, & \text{if } f(\mathbf{x}) < \rho_0 \\ 1, & \text{else} \end{cases} \tag{10}$$

where $I(\mathbf{x})$ is an indicator function, and 0 and 1 represent the empty and the solid phase, respectively, and $p_0$ is the threshold value. Thus the final filling fraction $\phi = V_0/V = \langle I(\mathbf{x}) \rangle$ of structures is determined by the choice of $\rho_0$. While such systems might be physically difficult to realise, we are primarily interested in understanding the optical properties of disorder systems, and question about chemical synthesis of potential structures are beyond the scope of this paper.

For the FDTD simulation, the structures $f(\mathbf{x})$ were imported to USCF chimera[72] using *Imagic, MRC, DM and STAR file i/o -tool*[73] and converted to Wavefront OBJ files using a similar threshold scheme.

**Gaussian processes.** The GP1-2 models where synthesised spectrally using fast Fourier transform (FFT) techniques[74–77]

$$f(\mathbf{x}) = \mathbf{S} * \mathbf{N} = \text{FFT}^{-1}[\text{FFT}(\mathbf{S}) \cdot \text{FFT}(\mathbf{N})] \tag{11}$$

where $\mathbf{S} = [\mathbf{S}_1, ..., \mathbf{S}_N]$, $\mathbf{N} = [\mathbf{N}_1, ..., \mathbf{N}_N]$ are ($N \times N \times N$) arrays and

$$\mathbf{S}_k = \begin{bmatrix} s_{11k} & s_{12k} & \cdots & s_{1Nk} \\ s_{21k} & s_{22k} & \cdots & s_{2Nk} \\ \vdots & \vdots & \ddots & \vdots \\ s_{N1k} & s_{N2k} & \cdots & s_{NNk} \end{bmatrix}, \mathbf{N}_k = \begin{bmatrix} n_{11k} & n_{12k} & \cdots & n_{1Nk} \\ n_{21k} & n_{22k} & \cdots & n_{2Nk} \\ \vdots & \vdots & \ddots & \vdots \\ n_{N1k} & n_{N2k} & \cdots & n_{NNk} \end{bmatrix},$$

, $s_{ijk} = K(r_{ijk})$, $r^2 = (\frac{N}{2} - i)^2 + (\frac{N}{2} - j)^2 + (\frac{N}{2} - k)^2$, and $n_{ijk} \sim \mathcal{N}(0, 1)$ are identically and independently sampled from normal distribution.

**Phase-field simulations.** The Cahn–Hilliard model used for generating the spinodal decomposition structures SD1-3 is given by

$$\frac{\partial f}{\partial t} = \nabla^2 M \left[ \frac{\partial W(f)}{\partial f} - \frac{1}{2}\epsilon(\nabla f)^2 \right] \tag{12}$$

$$W(f) = Af^2(1 - f)^2 \tag{13}$$

where $M$ and $\epsilon$ are the mobility and the diffusion constants, and $W(f)$ is the mixing energy between the two equilibrium phases, and was numerically solved using semi-explicit spectral method[78,79]

$$\mathbf{F}^{[n+1]} = \frac{\mathbf{F}^{[n]} - \Delta t k^2 M W'(\mathbf{F}^{[n]})}{1 + \Delta t k^4 M \epsilon} \tag{14}$$

where $\mathbf{F}^{[i]} = \text{FFT}(\mathbf{f}^{[i]})$, with time step $\Delta t = 1 \times 10^{-3}$, $A = 1$, $M = 1$, and the coefficient $\epsilon$ was chosen from [0.04,1.00] to reach the desired length scale.

The FC1-5 models were created with Functionalized Cahn–Hilliard model[35,36]

$$\frac{\partial f}{\partial t} = \Delta \left[ E_b \left( \epsilon^2 \Delta - W_b''(f) \right) \left( \epsilon \Delta f - W_b'(f) \right) + \eta_h \epsilon^2 \Delta f - \eta_m W_s'(f) \right] \tag{15}$$

$$W_s(f) = W_b(f) = \frac{1}{2}(f + 1)^2 \left( \frac{1}{2}(f - 1)^2 + \frac{\tau}{3}(f - 2) \right). \tag{16}$$

using the CUDA code of Jones[37] with $\eta_h = 5$ and $\eta_m = [-7.25, -0.5, 3, 6, 10]$ for FC1-5 respectively.

**Synthesis of anisotropic structures.** Anisotropic structures where synthesis using a the method of Essery[80] where the mobility coefficient $\epsilon$ in Eqs. (12), and (15) was replaced by tensor

$$\epsilon = \begin{bmatrix} \epsilon_x & 0 & 0 \\ 0 & \epsilon_y & 0 \\ 0 & 0 & \epsilon_z \end{bmatrix} \tag{17}$$

and a similar method for the GP models by anisotropic scaling $r^2 = \epsilon_x^2(\frac{N}{2} - i)^2 + \epsilon_y^2(\frac{N}{2} - j)^2 + \epsilon_z^2(\frac{N}{2} - k)^2$ in Eq. (11).

Anisotropy values were calculated using the Karambola software package[50,51,81] from the STL converted files. Anisotropy values for beetle scales were calculated from the files `CY_cube.npy` and `LS_cube.npy` of Burg and coworkers' dataset[38].

**Calculation of two-point correlation functions.** To determine the characteristic length scale of the simulated structures $I(\mathbf{x})$, we used radially averaged 2-point correlation function calculated using a FFT method[82]

$$S_2(r) = \frac{\sum_{l,m,n \in \Omega} \text{FFT}^{-1} \left( |\text{FFT}(I(\mathbf{x}))|^2 \right)}{\omega} \tag{18}$$

where $\Omega = \{ (l, m, n) \,|\, l^2 + m^2 + n^2 = r^2, \, r \leq N/2 \}$ and $\omega$ is the number of elements in $\Omega$. In the anisotropy investigations $S_2(r)$ was calculated along axis parallel to the incoming wave $\coloneqq \langle I(\mathbf{x})I(\mathbf{x} + l) \rangle$, $l = r$.

The correlation length, $l_c$, was defined to be the width of the slope of the 2-point correlation function, which in the case oscillating $S_2(r)$ coincides with first minima $\arg\min_r S_2(r)$, and was determined by fitting Corson's formula[28,83] (see Supplementary Fig. S1)

$$\begin{aligned} C_2(r) &= \phi^2 + \phi(1 - \phi)e^{-cr^n} \\ &= \phi^2 + a(\phi - \phi^2) \end{aligned} \tag{19}$$

to $S_2(r)$, and then setting

$$l_c \coloneqq \sqrt[n]{\frac{\log(a)}{-c}} \Bigg|_{a = 0.007} \tag{20}$$

i.e. the distance where the drop, $a = e^{-cr^n}$, in value of $C_2(r)$ was 0.007.

Correlation strength was defined as the relative depth of the minima of $S_2(r)$

$$f_{\min} = \min \left( \frac{S_2(r) - \phi^2}{\phi - \phi^2} \right) \tag{21}$$

**Calculation of Minkowski measures.** The scalar Minkowski functionals, Eqs. (3)–(6), and tensor Eq. (8) were calculated with Karambola[50,51,81] from the OBJ converted files.

**FDTD simulations.** FDTD simulations were performed using Lumerical 2020a-r5 (Ansys Canada Ltd.). In each simulation, a cubic structure was illuminated at normal incidence with a linear polarised plane wave across a broad spectral range

($\lambda = 300-800$ nm), with periodic boundary conditions in the perpendicular directions. The two-phase structures were assigned refractive indices $n_1 = 1$, $n_2 = 1.5$ unless otherwise stated. The numerical stability and convergence was ensured with the adequate boundary condition, and the simulations were carried out until all incoming light had either reflected or transmitted. For reflectance measurements, the back-scattered light was collected by a reflectance monitor above the source and summed over the monitor area to obtain the total reflectance spectra $R(\lambda)$ (Fig. 1b). The simulated spectra are therefore comparable to experimental reflectance spectra obtained using an integrating sphere. The mean integrated reflectance $\langle R \rangle$ was then obtained by spectral averaging of $R(\lambda)$ over the $\lambda = [300, 800]$ nm range. For the mean free path calculations, the ballistic transmission was recorded using additional TM and TE monitors.

## Data availability

Raw spectra and additional data related to this publication is available at the University of Cambridge data repository[84].

## Code availability

Code for generating the GP1-2 and SD1-3 structures, $S_2(r)$ and Minkowski function calculations are available at the University of Cambridge data repository[84].

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

## Acknowledgements

The authors thank Prof. Rémi Carminati for his helpful suggestions on the manuscript. J.S.H. is grateful for financial support from the Emil Aaltonen Foundation and Academy of Finland grant (no. 347789). L.S. acknowledges the support of the Isaac Newton Trust and the Swiss National Science Foundation under project 40B1-0_198708. T.G.P. acknowledges the EPSRC NanoDTC, project EP/L015978/1 and EP/T517847/1. This work is part of a project that has received funding from the European Union's Horizon 2020 research and innovation programme under the Marie Skłodowska-Curie grant agreement No. 893136 and the ERC SeSaME ERC-2014-STG H2020 639088. The FDTD simulations in this work were performed using resources provided by the Cambridge Service for Data Driven Discovery (CSD3) operated by the University of Cambridge Research Computing Service (www.csd3.cam.ac.uk), provided by Dell EMC and Intel using Tier-2 funding from the Engineering and Physical Sciences Research Council (capital grant EP/P020259/1), and DiRAC funding from the Science and Technology Facilities Council (www.dirac.ac.uk). The in silico synthesis of the FC1-5 structures were performed using computer resources provided by the Aalto University School of Science "Science-IT" project (https://scicomp.aalto.fi/).

## Author contributions

J.S.H performed the in silico synthesis and data analysis, G.J. and J.S.H. carried out the FDTD simulations, T.G.P. performed calculations of the theoretical optimal filling fraction, T.G.P. and J.S.H. wrote the manuscript, J.S.H., G.J., T.G.P., L.S., and S.V. designed the experiments and commented on results.

## Competing interests

The authors declare no competing interests.
