## [Peer Review file · Communications Physics]

Reviewers' comments:

Reviewer #1 (Remarks to the Author):

This paper performs computational investigation on optical reflectance of various random nanostructures and finds that, for isotropic structures, filling fraction and correlation length are two dominant factors that determine the reflectance. More minor but not negligible factor is identified as the surface-averaged mean curvature. Further, it shows that structural anisotropy can be optimized to achieve maximum reflectance. The authors did an extensive study on 10 different structure classes generated by the Gaussian process, the Cahn-Hilliard equation, and the functionalized Cahn-Hilliard equation. This is a welcome study going beyond the kinds of computational structures in the past. The results of optical simulations for all these structures are ample and well-presented in the figures. Many of the results are also included in the Supporting Information, which is helpful for readers. However, a major conclusion is not supported by the simulation results, some results are obtained by unacceptable approaches, and the conclusions remain skeptical due to lack of information on confidence intervals. When these points are corrected, I would recommend publication of the manuscript. Major points: 1. The authors claim that the type of disordered morphology (i.e., colloidal, network, cellular) is ultimately irrelevant as any structural class can give “comparable” scattering efficiency after optimization. This claim is made in Abstract as a major conclusion and is repeated throughout the manuscript. Let’s estimate what the word “comparable” means quantitatively. The optimized reflectance for sphere and FC5 is 0.58 and 0.46, respectively, in Fig. S6. When scattering strength ($1/lltt$) is calculated based on Eq. S9, the reflectance values give $1/lltt = 0.36 \mu\mu\mu\mu^{-1}$ and $0.59 \mu\mu\mu\mu^{-1}$, respectively (this estimation is only approximate as described in the next point). Thus, the sphere structure has 1.6 times stronger scattering strength than FC5. Such a large difference cannot be said “comparable”. The authors make such a claim just based on reflectance. However, scattering strength or scattering efficiency is not directly proportional to reflectance. Basically, the authors are making claims on an intrinsic property based on a thickness-dependent quantity. 2. While only reflectance is presented for optical properties in the figures of the manuscript, the mean free paths are calculated in Supporting Information and discussed in the main text. However, the method used in calculating the transport mean free path ($lltt$) is unacceptable, so that the $lltt$ results are questionable. Equation S9 is valid only for $LL \gg lltt$. The authors used $LL = 0.5, 1, \dots, 5 \mu\mu\mu\mu$ and the obtained $lltt$ ranges from $2 \mu\mu\mu\mu$ to $4 \mu\mu\mu\mu$, so that Eq. S9 becomes invalid for the sampled structures as even $LL < lltt$ for some thin structures. Moreover, Eq. S9 should be used for isotropic media. Figure S10 is meaningless especially for highly anisotropic structures ($\alpha \rightarrow 1$). 3. For a particular structure class with a $(\phi\phi, llcc)$ combination in Fig. 3, only one structure is used to calculate reflectance. Without knowing how much one particular structure can represent the structure class with a $(\phi\phi, llcc)$ combination, we cannot draw a reliable conclusion. In Ref. [20] Fig. 3g, SD structures of a size $20 \times 25 \times 5 \mu\mu\mu\mu^3$ show an absolute error of $\pm 3\%$ in spectral average reflectance at a 30% volume fraction. However, the authors used a much smaller volume of $5 \times 5 \times 5 \mu\mu\mu\mu^3$, so that the error could be significantly greater. Thus, confidence intervals need to be indicated in separate figures (as in Fig. S8) and given in Supporting Information, so that any conclusions from the figure should be supported. Also, it should be stated how the confidence intervals are determined (Fig. S8 caption doesn’t say this), such as the number of structures simulated and the initial filling fractions varied in the structure generation. Addressing this point might require considerable computation. Minor points: 1. In Introduction, $(n_1 n_2)/n_1$ should be corrected to $(n_1 - n_2)/n_1$. 2. Because the Minkowski functionals in Eqs. (3)-(6) scales with the structure volume or surface area, specific quantities such as volume fraction and

specific surface could be used instead. If the authors have justification that Eqs. (3)-(6) are preferred to the specific quantities, this comment doesn't need to be followed. However, if the authors want to stick to Eqs. (3)-(6), the units for V_1 and V_2 are missing in Fig. 2c. Figure 4 should also include units. 3. In spectral averaging for mean reflectance, what is the range of spectrum? 4. In page 6, $R \sim \lambda^4$ is not correct. Did the author mean $R \sim 1/\lambda^4$? But still this is not true because only the scattered intensity is proportional to $1/\lambda^4$, and we do not know what happens to R for multiple scattering. This error seems to be a result of the authors' confusion between intrinsic properties and a thickness-dependent quantity as described in my major point 1. 5. In page 6, the authors compare the second-order approximation in Ref. [24] with their results. Specifically, the scattering mean free path is the shortest at filling fraction of 50% in the approximation, while their results show optimal filling fraction well below 50%. This discussion confuses scattering mean free path and transport mean free path. Transport mean free path includes the effect of asymmetry factor but scattering mean free path doesn't. Their optimal filling fraction minimized the transport mean free path, but not necessarily the scattering mean free path. 6. In Ref. [20] Fig. 3e, SD structure with a 30% filling fraction shows a higher reflectance than a beetle scale structure (Cyphochilus). Figure 5 and Fig. S8 in the present manuscript do not seem to agree with this: The beetle structure looks similar to FC3 or FC4 with an anisotropy $\alpha = 0.4$ but these structures show a quite similar reflectance to SD1, SD2, and SD3 at $\alpha = 0$. As the manuscript refers to the beetle scale structure in several places, the authors are advised to comment on this seeming disagreement with Ref. [20].

Reviewer #3 (Remarks to the Author):

The authors present in their manuscript a unifying approach to designing and characterizing the expected whiteness of certain structures. This is an important contribution to the field, as many different structures made from many different materials have recently been identified to deliver brilliant whiteness. While few papers so far presented models behind the whiteness to actually guide the design of these structures, the authors rigorously analyze the different structure types based on topological parameters. Thus, they are able to explain, why seemingly completely different structures are able to deliver the same degree of whiteness. To achieve this the authors employ Minkowski functional to analyze the morphology and find out that especially two quantities are of importance: the integral mean curvature and the surface area.

The paper is written in a very clear and concise way so that the line of the story can easily be followed. Especially the many numerical examples support the understanding of the complex mathematics behind it. The manuscript can thus be published in its present form. There are only minor points that might be taken into account:

1. As far as I understand, the analysis has been done on a set of structures with preselected parameters. This is great to develop an intuition for optimal structural design. However, would it be possible to perform a principal component analysis taking into account all parameters to even better classify the structures, and could this be used to perhaps even automatically optimize the structures with respect to certain experimental constraints like index contrast, feature size, and so on?
2. There is a small typo on page 6, second last paragraph before "The role of interfacial curvature": It should read "With increasing contrast,..." instead of "With increase contrast..."

In conclusion, a really very interesting paper that will guide the discussion about optimal design for brilliant whiteness. It should definitely be published in Communication Physics.

(Note: Due to technical delay by the journal we did not get access to reviewer comments until 5th of January 2023)

We thank whole heartedly both reviewers for their recognition and critical comments regarding our manuscript.

Reviewer 1 comments & reply

“

”This paper performs computational investigation on optical reflectance of various random nanostructures and finds that, for isotropic structures, filling fraction and correlation length are two dominant factors that determine the reflectance. More minor but not negligible factor is identified as the surface-averaged mean curvature. Further, it shows that structural anisotropy can be optimized to achieve maximum reflectance.

The authors did an extensive study on 10 different structure classes generated by the Gaussian process, the Cahn-Hilliard equation, and the functionalized Cahn-Hilliard equation. This is a welcome study going beyond the kinds of computational structures in the past. The results of optical simulations for all these structures are ample and well-presented in the figures. Many of the results are also included in the Supporting Information, which is helpful for readers. However, a major conclusion is not supported by the simulation results, some results are obtained by unacceptable approaches, and the conclusions remain skeptical due to lack of information on confidence intervals. When these points are corrected, I would recommend publication of the manuscript.

”

“

Major points:

1. The authors claim that the type of disordered morphology (i.e., colloidal, network, cellular) is ultimately irrelevant as any structural class can give “comparable” scattering efficiency after optimization. This claim is made in Abstract as a major conclusion and is repeated throughout the manuscript. Let’s estimate what the word “comparable” means quantitatively. The optimized reflectance for sphere and FC5 is 0.58 and 0.46, respectively, in Fig. S6. When scattering strength ($1/l_t$) is calculated based on Eq. S9, the reflectance values give $1/l_t = 0.36 \mu\text{m}^{-1}$ and $0.59 \mu\text{m}^{-1}$, respectively (this estimation is only approximate as described in the next point). Thus, the sphere structure has 1.6 times stronger scattering strength than FC5. Such a large difference cannot be said “comparable”. The authors make such a claim just based on reflectance. However, scattering strength or scattering efficiency is not directly proportional to reflectance. Basically, the authors are making claims on an intrinsic property based on a thickness-dependent quantity.

”

We thank the reviewer for raising the criticism of our qualitative use of the word “comparable”. We would like to note that

1.1) we have already acknowledged in the original manuscript (MS) that FC1 has the best performance, and that FC5 is an exception¹. For all others structures, a set of parameters can be found resulting to reflectance between 0.5 – 0.55, which leads to a scattering strength difference smaller than factor 1.5 which in our opinion is ‘comparable’

1.2) In the abstract, which the reviewer also refers to, we do indeed note that optical scattering efficiency can be obtained from broad range of disordered structures, but we also emphasize “...*especially when structural anisotropy is included.*” Therefore making quantitative comparison between colloidal spherical structures and FC5 from figure S6 is a bit of a misrepresentation of our claim, especially since we show in Figure 5 and fig. S7 the effect of further anisotropy optimisation that enables to obtain a reflectance $R = 0.60$ for FC5 as shown in Figure fig. S7.

Regarding intrinsic properties vs thickness dependent quantities, we address the former on the reply on the next major point. Regarding the latter, the conclusion can already be drawn from reflectance, given that all simulations are performed with the exact same thickness (which would be difficult to achieve with experimental samples), $L^3 = 5 \times 5 \times 5 \mu\text{m}^3$ therefore eliminating this parameter.

¹p. 5: “with the notable exception of the ‘cellular FC5 class at low filling fraction...”, p. 8 “The exception was the ‘cellular FC5 structural class, whose structures showed increased reflectance at high anisotropy...”

Figure S7: "Example of reflectance optimization of highly anisotropic FC5 by tuning the correlation length."

“

2. While only reflectance is presented for optical properties in the figures of the manuscript, the mean free paths are calculated in Supporting Information and discussed in the main text. However, the method used in calculating the transport mean free path (l_t) is unacceptable, so that the l_t results are questionable. Equation S9 is valid only for $L \gg l_t$. The authors used $L = 0.5, 1, \dots, 5 \mu\text{m}$ and the obtained l_t ranges from $2 \mu\text{m}$ to $4 \mu\text{m}$, so that Eq. S9 becomes invalid for the sampled structures as even $L < l_t$ for some thin structures. Moreover, Eq. S9 should be used for isotropic media. Figure S10 is meaningless especially for highly anisotropic structures ($\alpha \rightarrow 1$).

”

The reviewer is of course right here that in general the diffusion theory is only valid for isotropic media with optical thickness $L/l_t > 8$ [R1, R2]. There are some independent literature that derive the transport mean free path for anisotropic case [R3]. Additionally Lee et al [R4] note that the validity of diffusion theory also depends on the nature of the scattering [R5–R7] and the thickness limit is lowered as the index contrast decreases [R8, R9] even down to cases where $L \ll l_t$ [R10]. To demonstrate this, we carry out an extended simulation for the GP1 structure between $0.5\text{--}30 \mu\text{m}$ (see Figure R1b). By using the anisotropic diffusion model [R3, R4]

$$T_b = e^{-L/l_{s,zz}} \quad (1)$$

$$T_{tot} = \frac{[K_{zz} + z_e] - [K_{zz} - z_e] \exp(-l/l_{s,zz})}{L/l_{t,zz} + 2z_e} c \quad (2)$$

where $l_{[*],zz}$ are zz -components of the anisotropic scattering and transport mean free path tensors respec-

tively, we observe excellent linear fit in the full range of $L = 0.5\text{-}30\ \mu\text{m}$, similar that to the results of Lee and coworkers [R4].

Therefore we do not consider our original results unacceptable nor meaningless, nor that redoing all the simulation with higher thickness would yield different results. However we do agree that it would be more appropriate, given the fact that we extend our investigation to anisotropic structures, to speak instead about anisotropic mean free paths in the manuscript. Therefore we have replace the isotropic mean free paths l_s, l_t with the anisotropic ones $l_{s,zz}, l_{t,zz}$, fitted using eqs. (1) and (2) (figures S10 and S11 are revised), and added an appropriate discussion to the manuscript highlighting the limitations of applying transport mean free path calculations to optically thin and anisotropic structures.

Figure R1: Transport mean free path $l_{t,zz}$ regression for the structures with $l_c = 300\ \text{nm}$, $V_0/V = 30\%$, $\alpha \approx 0$. **a-b)** To test the independence of $l_{t,zz}$ on sample thickness L , the GP1 structure thickness is extend from $0.5\text{-}5\ \mu\text{m}$ to $0.5\text{-}30\ \mu\text{m}$ range and shows an excellent linear fit for anisotropic diffusion model $y = -2z_e + K_{zz} \frac{1 - \exp(-L/l_{s,zz})}{T} + z_e \frac{1 + \exp(-L/l_{s,zz})}{T}$. **c)** The same regression for the remaining structures on the original $L = 0.5\text{-}5\ \mu\text{m}$ range.

“

3. For a particular structure class with a (ϕ, l_c) combination in Fig. 3, only one structure is used to calculate reflectance. Without knowing how much one particular structure can represent the structure class with a (ϕ, l_c) combination, we cannot draw a reliable conclusion. In Ref. [R11] Fig. 3g, SD structures of a size $20 \times 25 \times 5 \mu\text{m}^3$ show an absolute error of $\pm 3\%$ in spectral average reflectance at a 30% volume fraction. However, the authors used a much smaller volume of $5 \times 5 \times 5 \mu\text{m}^3$, so that the error could be significantly greater. Thus, confidence intervals need to be indicated in separate figures (as in Fig. S8) and given in Supporting Information, so that any conclusions from the figure should be supported. Also, it should be stated how the confidence intervals are determined (Fig. S8 caption doesn't say this), such as the number of structures simulated and the initial filling fractions varied in the structure generation. Addressing this point might require considerable computation.

”

Question about the reflectance confidence intervals regarding simulation volumes is very valid one. In general different investigations use typically thicknesses between 3 [R12, R13] to 7 μm [R14, R15], and lateral dimension from 2-7 μm [R4, R16], depending on boundary conditions (PML or PBC), so our choice of $5 \times 5 \times 5 \mu\text{m}^3$ is not untypical.

Moreover reliability of our results are not inferior to that of [20] (a.k.a [R11]). Although the authors of Ref. [R11] do excellent spinodal decomposition and FDTD simulations, the used $20 \times 25 \times 5 \mu\text{m}^3$ volumes are limited by the observation that they are obtained by tessellating subvolumes of sizes between $\sim 4 \times 4 \times 4 \mu\text{m}^3$ and $\sim 13 \times 13 \times 5 \mu\text{m}^3$ to obtain $20 \times 25 \times 5 \mu\text{m}^3$ slabs, as evidenced from the [R11, Fig.4a-d)], cf. Figure R2. Such repetition of the same disordered pattern laterally is not in practice different from our approach of using $5 \times 5 \times 5 \mu\text{m}^3$ volumes with periodic boundary conditions in the lateral direction.

It's of course reasonable to present estimate of spectral confidence intervals (e.g. standard deviations), for instance in the same as manner as in Fig. 3g in [R11], but in our case this becomes redundant as the full information (raw data) on spectral variation is already available in Fig. S3 (which shows very similar trends as in [R11, Fig. 4f and Fig. 3e-f]). That is, the spectral variation decreases as the correlation length and filling fraction increase (cf. Figure R3). For instance the *“ $\pm 3\%$ in spectral average reflectance at a 30% volume fraction”* is true for large correlation lengths in [R11], as well for our case, and likewise when moving towards Rayleigh limit (shorter correlation lengths) the spectral variation is greater in both investigations (cf. Fig. 4f in [R11] and Fig. R2).

Thus, given effectively the same simulation volume size and spectral variation we are not in an inferior position in comparison to [R11] to make reliable conclusions from our simulations.

Regarding Fig. S8, the confidence intervals are related to predictive Gaussian Process interpolations, as noted in the reference [R18, eq. (2.20-21)] given in the caption, where the parameters l_2, σ_n control the rigidity of the interpolating curves and tightness of the fit, and are not to be confused with spectral variation. We have further clarified this in the revised ESI.

Fig. 4 Spectral response versus spinodal characteristic length scale. Representative 2D cross sections in the xy plane for a series of spinodal slabs of varying characteristic length scale **a** 222 nm, **b** 325 nm, **c** 500 nm and **d** 690 nm. Each slice, regardless of length scale is $5 \mu\text{m} \times 20 \mu\text{m}$ and has a filling fraction of 30% white pixels. **e** $G_r(x)$ results for the 30:70 spinodal filling fraction for four different characteristic length scales. The dashed lines show the length scales of the *Cyphochilus* and the *L. stigma* beetles. **f** FDTD reflectance results for $5 \mu\text{m} \times 20 \mu\text{m}$ 30:70 spinodal optical structures having different characteristic correlation lengths

Figure R2: From [20] (a.k.a. Burg et al [R11]), red lines are added to highlighting the tessellation.

Figure R3: Comparison of full reflectance spectra of selected correlation length of **a-b)** SD1, SD2 (from figure S3) and **d)** [R11, Fig. 4f] (re plotted using WebPlotDigitizer [R17]). Note that in the first curve the correlation length l_c is smaller in **a-b)** than in **c)**, and the small l_c leads towards Rayleigh type behaviour.

“

Minor points:

1. In Introduction, $(n_2 n_1) / n_1$ should be corrected to $(n_2 - n_1) / n_1$.

”

This has been corrected.

“

2. Because the Minkowski functionals in Eqs. (3)-(6) scales with the structure volume or surface area, specific quantities such as volume fraction and specific surface could be used instead. If the authors have justification that Eqs. (3)-(6) are preferred to the specific quantities, this comment doesn't need to be followed. However, if the authors want to stick to Eqs. (3)-(6), the units for V_1 and V_2 are missing in Fig. 2c. Figure 4 should also include units.

”

Units were originally in voxels, and have now been converted to SI units.

“

3. In spectral averaging for mean reflectance, what is the range of spectrum?

”

Our spectral averaging is carried between [300,800]nm, and has now been specified in the manuscript.

“

4. In page 6, $R \sim \lambda^4$ is not correct. Did the author mean $R \sim 1/\lambda^4$? But still this is not true because only the scattered intensity is proportional to $1/\lambda^4$, and we do not know what happens to R for multiple scattering. This error seems to be a result of the authors' confusion between intrinsic properties and a thickness-dependent quantity as described in my major point 1.

”

The reviewer is correct about the typo, it should read $R \sim \lambda^{-4}$. But this is a mere observation that the spectral response of structures at small correlation lengths ($l_c \leq 200$ nm) has a Rayleigh type dependency.

Even if the term Rayleigh dependency is historically used for single scattering classification, we believe that in our main text, ("*...structures start to exhibit Rayleigh type behaviour*"), it is clear that we are referring to structures where we have multiple scattering.

“

5. In page 6, the authors compare the second-order approximation in Ref. [24] with their results. Specifically, the scattering mean free path is the shortest at filling fraction of 50% in the approximation, while their results show optimal filling fraction well below 50%. This discussion confuses scattering mean free path and transport mean free path. Transport mean free path includes the effect of asymmetry factor but scattering mean free path doesn't. Their optimal filling fraction minimized the transport mean free path, but not necessarily the scattering mean free path.

”

The reviewer is correct that the transport mean free path is the relevant parameter in general, but in the independent scattering approximation the scattering and the transport mean free paths are equal, so the initial approach is correct, albeit a crude estimate that predicts the optimal filling fraction reasonably well. We have amended the text to emphasise the crudeness of this estimation method. Then, to explore the effect of structural correlations, we applied equations from Vynck et al [24] ([R19]) to idealised two phase structures to see how l_s and l_t depends on filling fraction and index contrast. We find trends in good agreement with the optimal filling fraction from the FDTD simulations (cf. Fig. S4 in the revised ESI).

“

6. In Ref. [20] Fig. 3e, SD structure with a 30% filling fraction shows a higher reflectance than a beetle scale structure (*Cyphochilus*). Figure 5 and Fig. S8 in the present manuscript do not seem to agree with this: The beetle structure looks similar to FC3 or FC4 with an anisotropy $\alpha = 0.4$ but these structures show a quite similar reflectance to SD1, SD2, and SD3 at $\alpha = 0$. As the manuscript refers to the beetle scale structure in several places, the authors are advised to comment on this seeming disagreement with Ref. [20].

”

We can compare our SD2 structure to *Cyphochilus* structure (CY) thanks to the beetle dataset released to the public domain by the authors of [R11, R20]. We do observe, similarly to [R11, fig. 3e], that SD2 structure shows higher reflectance than beetle scales, except when $\lambda > 600$ nm, see Figure R4. However the reflectance of SD2 is expected to be even higher, if the correlation lengths for SD2 and CY were matched (300 nm vs. 325 nm), given that reflectance increases for SD2 from $l_c = 300$ nm \rightarrow 400 nm (cf. fig. 3c). Also one should note that we use periodic boundary conditions in xy-direction in Lumerical in comparison to [R11], who use perfectly matched (absorbing) layer in xy-direction. Thus our Lumerical setup corresponds to integrating sphere setup, where as in [R11] light is collected from more limited angle that depends on the monitor distance from the sample, which is not specified in [R11].

Figure R4: Comparison of 30 % SD2 structure to *Cyphochilus* simulated from dataset from [R20].

Reviewer 2 comments & reply

“

The authors present in their manuscript a unifying approach to designing and characterizing the expected whiteness of certain structures. This is an important contribution to the field, as many different structures made from many different materials have recently been identified to deliver brilliant whiteness. While few papers so far presented models behind the whiteness to actually guide the design of these structures, the authors rigorously analyze the different structure types based on topological parameters. Thus, they are able to explain, why seemingly completely different structures are able to deliver the same degree of whiteness. To achieve this the authors employ Minkowski functional to analyze the morphology and find out that especially two quantities are of importance: the integral mean curvature and the surface area.

The paper is written in a very clear and concise way so that the line of the story can easily be followed. Especially the many numerical examples support the understanding of the complex mathematics behind it. The manuscript can thus be published in its present form.

”

We do thank the Reviewer for the very kind evaluation of our manuscript.

“

There are only minor points that might be taken into account:

1. As far as I understand, the analysis has been done on a set of structures with preselected parameters. This is great to develop an intuition for optimal structural design. However, would it be possible to perform a principal component analysis taking into account all parameters to even better classify the structures, and could this be used to perhaps even automatically optimize the structures with respect to certain experimental constraints like index contrast, feature size, and so on?

”

The reviewer is correct in that we do line searches in the parameter space. We can certainly try PCA classification of the structures based on the collected parameters, see fig. R5. However in our case PCA is not particularly helpful for improving the classifying the structures.

Figure R5: Principal component analysis (PCA), done separately for each correlation length for convenience.

However in a hypothetical case where PCA would yield clearly distinct structural classes, the idea of automatic optimization would certainly be very cool, but probably not trivial to implement.

For instance the used models like Cahn-Hilliard and Functionalized Cahn-Hilliard have their own predefined energy functionals $E(f)$, (which are independent of PCA results ("energy")), and lead to specific disordered

structures $f^{(*)}$ under energy minimization

$$\operatorname{argmin}_f E(f)$$

To utilize the PCA results in the same manner efficiently one would have to find a clever energy function $E_{[?]}(f)$ that favour features suggested by the PCA results, and an efficient solver (e.g. gradient optimizer) to minimize that energy.

It's more likely that one would need to resort to Markov Chain Monte Carlo optimization where one randomly alters a pixel (voxel) in the simulation volume, calculates the energy of the new configuration by determining the all the necessary parameter values using auxiliary programs like Karambola ($V_1, V_2, V_3, \alpha, \dots$) etc, and calculating their least square mean difference to optimal values predicted by PCA. Given that typical simulation volumes are 100^3 voxels in size, this would probably have a very low speed of convergence, and since the energy needs to be calculated using auxiliary programs in each step, the computational time might not be very feasible.

These are definitely interesting research topics though. However given the fact that our take away message is that very different systems can be optimized by just tuning a handful of parameters (l_c, V_0, α), finding and implementing automated optimizer would be more suitable for future studies.

Finally, we would like to one more time thank both reviewers for reading and commenting our manuscript, and hope that we have been able to answer the critique and questions satisfyingly.

On the behalf of the authors,
Professor Silvia Vignolini

References

- [R1] K. M. YOO, F. LIU, and R. R. ALFANO. When does the diffusion approximation fail to describe photon transport in random media? *Phys. Rev. Lett.* **Mar. 1990**, *64*, pp. 2647–2650. DOI: [10.1103/PhysRevLett.64.2647](https://doi.org/10.1103/PhysRevLett.64.2647).
- [R2] G. MAZZAMUTO, L. PATTELLI, C. TONINELLI, and D. S. WIERSMA. Deducing effective light transport parameters in optically thin systems. *New Journal of Physics.* **Feb. 2016**, *18*, 2, p. 023036. DOI: [10.1088/1367-2630/18/2/023036](https://doi.org/10.1088/1367-2630/18/2/023036).
- [R3] S. E. HAN. Transport mean free path tensor and anisotropy tensor in anisotropic diffusion equation for optical media. *Journal of Optics.* **June 2020**, *22*, 7, p. 075606. DOI: [10.1088/2040-8986/ab954d](https://doi.org/10.1088/2040-8986/ab954d).
- [R4] S. H. LEE, S. M. HAN, and S. E. HAN. Anisotropic diffusion in Cyphochilus white beetle scales. *APL Photonics.* **2020**, *5*, 5, p. 056103. DOI: [10.1063/1.5144688](https://doi.org/10.1063/1.5144688).
- [R5] R. H. J. KOP, P. DE VRIES, R. SPRIK, and A. LAGENDIJK. Observation of Anomalous Transport of Strongly Multiple Scattered Light in Thin Disordered Slabs. *Phys. Rev. Lett.* **Dec. 1997**, *79*, pp. 4369–4372. DOI: [10.1103/PhysRevLett.79.4369](https://doi.org/10.1103/PhysRevLett.79.4369).
- [R6] X. ZHANG and Z.-Q. ZHANG. Wave transport through thin slabs of random media with internal reflection: Ballistic to diffusive transition. *Phys. Rev. E.* **July 2002**, *66*, p. 016612. DOI: [10.1103/PhysRevE.66.016612](https://doi.org/10.1103/PhysRevE.66.016612).
- [R7] I. M. VELLEKOOP, P. LODAHL, and A. LAGENDIJK. Determination of the diffusion constant using phase-sensitive measurements. *Phys. Rev. E.* **May 2005**, *71*, p. 056604. DOI: [10.1103/PhysRevE.71.056604](https://doi.org/10.1103/PhysRevE.71.056604).
- [R8] R. ELALOUI, R. CARMINATI, and J.-J. GREFFET. Diffusive-to-ballistic transition in dynamic light transmission through thin scattering slabs: a radiative transfer approach. *J. Opt. Soc. Am. A.* **Aug. 2004**, *21*, 8, pp. 1430–1437. DOI: [10.1364/JOSAA.21.001430](https://doi.org/10.1364/JOSAA.21.001430).
- [R9] L. PATTELLI, G. MAZZAMUTO, D. S. WIERSMA, and C. TONINELLI. Diffusive light transport in semitransparent media. *Phys. Rev. A.* **Oct. 2016**, *94*, p. 043846. DOI: [10.1103/PhysRevA.94.043846](https://doi.org/10.1103/PhysRevA.94.043846).
- [R10] D. J. DURIAN. Influence of boundary reflection and refraction on diffusive photon transport. *Phys. Rev. E.* **Aug. 1994**, *50*, pp. 857–866. DOI: [10.1103/PhysRevE.50.857](https://doi.org/10.1103/PhysRevE.50.857).
- [R11] S. L. BURG, A. WASHINGTON, D. M. COLES, A. BIANCO, D. MCLOUGHAN, O. O. MYKHAYLYK, J. VILANOVA, A. J. C. DENNISON, C. J. HILL, P. VUKUSIC, S. DOAK, S. J. MARTIN, M. HUTCHINGS, S. R. PARNELL, C. VASILEV, N. CLARKE, A. J. RYAN, W. FURNASS, M. CROUCHER, R. M. DALGLIESH, S. PREVOST, R. DATTANI, A. PARKER, R. A. L. JONES, J. P. A. FAIRCLOUGH, and A. J. PARNELL. Liquid-liquid phase separation morphologies in ultra-white beetle scales and a synthetic equivalent. *Communications Chemistry.* **Aug. 2019**, *2*, 1, p. 100. ISSN: 2399-3669. URL: <https://doi.org/10.1038/s42004-019-0202-8>.
- [R12] L. PATTELLI, A. EGEL, U. LEMMER, and D. S. WIERSMA. Role of packing density and spatial correlations in strongly scattering 3D systems. *Optica.* **Sept. 2018**, *5*, 9, pp. 1037–1045. DOI: [10.1364/OPTICA.5.001037](https://doi.org/10.1364/OPTICA.5.001037).
- [R13] K. DJEGHDI, U. STEINER, and B. D. WILTS. 3D Tomographic Analysis of the Order-Disorder Interplay in the Pachyrhynchus congestus mirabilis Weevil. *Advanced Science.* **2022**, *9*, 26, p. 2202145. DOI: [10.1002/advs.202202145](https://doi.org/10.1002/advs.202202145).
- [R14] D. T. MEIERS, M.-C. HEEP, and G. VON FREYMAN. Invited Article: Bragg stacks with tailored disorder create brilliant whiteness. *APL Photonics.* **2018**, *3*, 10, p. 100802. DOI: [10.1063/1.5048194](https://doi.org/10.1063/1.5048194).
- [R15] F. UTEL, L. CORTESE, D. S. WIERSMA, and L. PATTELLI. Optimized White Reflectance in Photonic-Network Structures. *Advanced Optical Materials.* **2019**, *0*, 0, p. 1900043. DOI: [10.1002/adom.201900043](https://doi.org/10.1002/adom.201900043).
- [R16] S. H. LEE, S. M. HAN, and S. E. HAN. Nanostructure regularity in white beetle scales for stability and strong optical scattering [Invited]. *Opt. Mater. Express.* **June 2021**, *11*, 6, pp. 1692–1704. DOI: [10.1364/OME.427047](https://doi.org/10.1364/OME.427047).

- [R17] A. ROHATGI. *Webplotdigitizer: Version 4.6*. 2022. URL: <https://automeris.io/WebPlotDigitizer>.
- [R18] C. E. RASMUSSEN and C. WILLIAMS. *Gaussian processes for machine learning*. MIT Press, 2006. URL: <http://www.gaussianprocess.org/gpml/>.
- [R19] K. VYNCK, R. PIERRAT, R. CARMINATI, L. S. FROUFE-PÉREZ, F. SCHEFFOLD, R. SAPIENZA, S. VIGNOLINI, and J. J. SÁENZ. *Light in correlated disordered media*. 2021. DOI: [10.48550/ARXIV.2106.13892](https://doi.org/10.48550/ARXIV.2106.13892).
- [R20] S. L. BURG, A. L. WASHINGTON, J. VILLANOVA, A. J. C. DENNISON, D. MCLOUGHLIN, O. O. MYKHAYLYK, P. VUKUSIC, W. FURNASS, R. A. L. JONES, A. J. PARNELL, and J. P. A. FAIRCLOUGH. X-ray nanotomography of complete scales from the ultra-white beetles *Lepidiota stigma* and *Cyphochilus*. *Scientific Data*. **May 2020**, 7, 1, p. 163. ISSN: 2052-4463. URL: <https://doi.org/10.1038/s41597-020-0502-y>.

Reviewer #1 (Remarks to the Author):

The authors have addressed my concerns in detail in the revised manuscript. I agree with the authors that reflectance at the same thickness can be used to compare scattering performance between different structures. The authors used a thickness that gives a middle range reflectance (20 - 60 %), so that a relative comparison is easy to see. But the use of reflectance loses the meaning of intrinsic properties. If the authors choose to use reflectance just for performance comparison, I will not object to their decision. But please modify or remove $R \sim \lambda^{-4}$, which is definitely incorrect.

Response to Referees

April 27, 2023

Reviewer 1 (Remarks to the Author):

“

The authors have addressed my concerns in detail in the revised manuscript. I agree with the authors that reflectance at the same thickness can be used to compare scattering performance between different structures. The authors used a thickness that gives a middle range reflectance (20 - 60 %), so that a relative comparison is easy to see. But the use of reflectance loses the meaning of intrinsic properties. If the authors choose to use reflectance just for performance comparison, I will not object to their decision. But please modify or remove $R \sim \lambda^{-4}$, which is definitely incorrect.

”

We are delighted to hear that we've address the Reviewer's concerns. We've removed the $R \sim \lambda^{-4}$ as requested by the reviewer. We thank the reviewers for their reading and commenting our manuscript and for their recommendation to accept the article.

On the behalf of the authors,
Professor Silvia Vignolini